# Hyperbaric oxygen protects HT22 cells and PC12 cells from damage caused by oxygen-glucose deprivation/reperfusion via the inhibition of Nrf2/System Xc-/GPX4 axis-mediated ferroptosis

Chunxia Chen[1☯]*, Wan Chen[2☯], Xing Zhou[1☯], Yaoxuan Li[3], Xiaorong Pan[4]*, Xiaoyu Chen[1]*

1 Department of Pharmacy, The People's Hospital of Guangxi Zhuang Autonomous Region & Guangxi Academy of Medical Sciences, Nanning, Guangxi, P. R. China, 2 Department of Emergency, The People's Hospital of Guangxi Zhuang Autonomous Region & Guangxi Academy of Medical Sciences, Nanning, Guangxi, P. R. China, 3 Department of Neurology, The People's Hospital of Guangxi Zhuang Autonomous Region & Guangxi Academy of Medical Sciences, Nanning, China, 4 Department of Hyperbaric Oxygen, The People's Hospital of Guangxi Zhuang Autonomous Region & Guangxi Academy of Medical Sciences, Nanning, Guangxi, P. R. China

☯ These authors contributed equally to this work.
* chunxia251401@126.com (CC); panxrong2015@163.com (XP); 1534746296@qq.com (XC)

**Data Availability Statement:** All relevant data are within the manuscript and its Supporting information files.

## Abstract

This study was to investigate the protective effect of hyperbaric oxygen (HBO) on HT22 and PC12 cell damage caused by oxygen-glucose deprivation/reperfusion-induced ferroptosis. A 2-h oxygen-glucose deprivation and 24-h reperfusion model on HT22 and PC12 cells was used to simulate cerebral ischemia-reperfusion injury. Cell viabilities were detected by Cell Counting Kit-8 (CCK-8) method. The levels of reactive oxygen species (ROS) and lipid reactive oxygen species (Lipid ROS) were detected by fluorescent probes Dihydroethidium (DHE) and C11 BODIPY 581/591. Iron Colorimetric Assay Kit, malondialdehyde (MDA) and glutathione (GSH) activity assay kits were used to detect intracellular iron ion, MDA and GSHcontent. Cell ferroptosis-related ultrastructures were visualized using transmission electron microscopy (TEM). Furthermore, PCR and Western blot analyses were used to detect the expressions of ferroptosis-related genes and proteins. After receiving oxygen-glucose deprivation/reperfusion, the viabilities of HT22 and PC12 cells were significantly decreased; ROS, Lipid ROS, iron ions and MDA accumulation occurred in the cells; GSH contents decreased; TEM showed that cells were ruptured and blebbed, mitochondria atrophied and became smaller, mitochondrial ridges were reduced or even disappeared, and apoptotic bodies appeared. And the expressions of Nrf2, SLC7A11 and GPX4 genes were reduced; the expressions of p-Nrf2/Nrf2, xCT and GPX4 proteins were reduced. Notably, these parameters were significantly reversed by HBO, indicating that HBO can protect HT22 cells and PC12 cells from damage caused by oxygen-glucosedeprivation/reperfusion via the inhibition of Nrf2/System Xc-/GPX4 axis-mediated ferroptosis.

**Funding:** This study was supported by the National Natural Science Foundation of China (81960246 and 81701089), the Guangxi Natural Science Foundation (2020GXNSFAA238003 and 2017GXNSFBA198010), the Guangxi Sanitation and Family Planning Committee Project (No. Z20201096) and the Guangxi Medical and Health Appropriate Technology Research and Development Project (S2020076).

**Competing interests:** The authors have declared that no competing interests exist.

## Introduction

Adequate cerebral blood perfusion is a key factor in maintaining normal brain function. chronic cerebral hypoperfusion (CCH) also known as chronic cerebral ischemia (CCI), prolonged ischemia of the brain can trigger neurodegeneration and eventually lead to progressive cognitive impairment. Currently, reperfusion is an effective treatment for patients with acute cerebral ischemia. However, reperfusion under severe cerebral ischemia is a double-edged sword, which often induces cerebral ischemia-reperfusion injury while restoring blood supply [1], which seriously threatens the life of patients. Recent studies have found that cell ferroptosis caused by abnormal iron metabolism may be an important pathophysiological mechanism of cerebral ischemia-reperfusion injury [2,3].

Ferroptosis was first proposed by Dixon et al. [4] in 2012. It is an iron-dependent, a new form of cell death caused by the formation of lipid peroxides. Ferroptosis is significantly different from apoptosis, necrosis, and autophagy in terms of morphology, biochemical characteristics, and gene expression [5,6]. In recent years, studies have found that ferroptosis is closely related to the occurrence and development of neurodegenerative diseases and traumatic nervous system injury diseases [7–10], and nerve cell death is the main pathological event of many neurological diseases [11–13]. Brain tissue is prone to attack by hydroxyl radicals due to the presence of a large amount of unsaturated fatty acids and low levels of antioxidant enzymes, such as glutathione peroxidase (GPX4) [14]. Therefore, neurological diseases are closely related to ferroptosis.

Hyperbaric oxygen (HBO) therapy is a clinical use of pure oxygen in the environment of 2–3 times atmospheric pressure to treat ischemic and hypoxic diseases, such as carbon monoxide poisoning, cerebral infarction, decompression sickness and coronary heart disease, etc., and has obtained certain curative effects. According to the US FDA and the International Society of Underwater Hyperbaric Medicine, 13 clinical indications including decompression sickness, carbon monoxide poisoning, gas embolism, acute craniocerebral injury, and post-radiotherapy tissue damage can be treated with HBO [15]. HBO treatment has a strong effect of reducing oxidative stress, a strong anti-inflammatory effect, protecting cholinergic nerves, and delaying neuronal apoptosis [16,17]. Our previous study [18] also found that hyperbaric oxygen can protect PC12 cells from damage caused by oxygen-glucose deprivation/reperfusion. However, whether this effect is related to the inhibition of ferroptosis remains unknown. Therefore, this study used the oxygen-glucose deprivation/reperfusion injury model of HT22 cells and PC12 cells to simulate cerebral ischemia-reperfusion in vitro to explore the effect of HBO on the ferroptosis pathway.

## Materials and methods

### Culture and treatment of cells

HT22 cells were obtained from American Type Culture Collection (ATCC, Maryland, USA) and cultured in medium consisting of Dulbecco's modified Eagle's medium (DMEM, Gibco, USA), 10% fetal bovine serum (FBS, Gibco, USA) and 1% penicillin-streptomycin (Solarbio, Beijing). PC12 cells (ATCC, Maryland, USA) were maintained in RPMI 1640 medium (Gibco, USA) supplemented with 10% horse serum (Merck, USA), 5% FBS and 1% penicillin-streptomycin. Cells were divided into control group, model group, Hyperbaric oxygen group (HBO) group and HBO+Ferrostatin-1 (HBO+F)group. Cells of the control group were untreated. Cells of the model group were cultured in medium without sugar and serum and then put into a 37°C incubator with a mixture of 95% $N_2$ and 5% $CO_2$ gas for 2 h for oxygen sugar deprivation. Then, the cells were cultured with complete medium in a 37°C incubator with a mixture

of 95% O2 and 5% CO2 for 24 h. Cells of the HBO group were placed in a hyperbaric chamber (Yantai Hongyuan CO., Ltd) and received pure oxygen (0.25 MPa, 60 min). Cells of the HBO +F group were received pure oxygen and then cultured in medium with Ferrostatin-1 (5 μmol/ L, ferroptosis inhibitor, MedChemExpresss, USA) for 24h. This study was conducted in accordance with the Declaration of Helsinki and approved by Ethics Committee of People's Hospital of Guangxi Zhuang Autonomous Region.

## Cell viability assay

Cell viability was determined using a Cell Counting Kit-8 (CCK8, Dojindo, Japan) assay. Briefly, after cells in each group received treatment accordingly, 10 μL of CCK-8 solution was added to each well and then the cells were incubated at 37°C in a 95% air/5% CO2 atmosphere for 1 h. The optical density of the plate was then measured at 450 nm by using a microplate reader.

## Detection of iron content in cells

Intracellular iron content was measured by Iron Colorimetric Assay Kit (APPLYGEN, Beijing). Cells in each group were received treatment accordingly and washed twice with cold PBS. 200 μl of Mixture A (prepared according to the instructions) was added to each well and the cells were incubated at a 60°C water bath for 1 h. 60 μl iron ion detection reagent was added to each well and the cells were incubated at room temperature for 30 min. The optical density of the plate was then measured at 570 nm by using a microplate reader.

## Detection of reactive oxygen species (ROS) in cells

Intracellular reactive oxygen species (ROS) level was measured by superoxide anion fluorescent probe (Dihydroethidium, DHE, Beyotime). Cells in each group were received treatment accordingly and washed twice with PBS. 1 ml of DHE diluted in DMEM was added to each well and the cells were incubated at 37°C in the dark for 20 min and then were washed with DMEM 3 times. The red fluorescence intensity was observed under a fluorescent inverted microscope and was calculated by Image J software.

## Measurement of lipid reactive oxygen species (Lip-ROS) in cells

Lipid ROS level was determined using BODIPYTM 581 /591 C11 dye (GLPBIO). Cells in each group were received treatment accordingly and washed twice with PBS. The cells were added with 10μmol /L BODIPYTM 581 /591 C11 and incubated at 37°C in the dark for 30 min and then were washed with PBS 3 times. The green fluorescence intensity was observed under a fluorescent inverted microscope and was calculated by Image J software.

## Detection of MDA and GSH content in cells

Cells in each group were received treatment accordingly. Then, individual levels of MDA and GSH in cells were measured using malondialdehyde (MDA) and glutathione (GSH) activity assay kits (Beyotime, China) respectively according to the Kit Instructions. Individual contents of MDA and GSH were measured at 450 and 405 nm, respectively, with a microplate reader.

## Observation of cell ultrastructure by transmission electron microscopy (TEM)

Cell ferroptosis-related ultrastructures were visualized using TEM. Briefly, cells were treated accordingly and fixed with cold 3% glutaraldehyde for 2 hours. Cells were then embedded in

**Table 1. PCR primer sequences.**

| Gene | Forward primer (5'-3') | Reverse primer (5'-3') |
| --- | --- | --- |
| Nrf2 | GTGGTTTAGGGCAGAAGG | TCTTTCTTACTCTGCCTCTA |
| SLC7A11 | GCATTCCCAGGGGCTAACAT | AATTTCTCCCATGCGGGTGT |
| GPX4 | CCGCTTATTGAAGCCAGCAC | TATCGGGCATGCAGATCGAC |
| GAPDH | CGTGTTCCTACCCCCAATGT | TGTCATCATACTTGGCAGGTTTCT |

1% osmium tetroxide and dehydrated. Subsequently, they were soaked in acetone and embedding medium overnight. Finally, they were stained with uranyl acetate and lead citrate. The ultrastructures of cells were examined using a transmission electron microscope (Hitachi H-7650, Japan).

### The relative quantitative real-time PCR analysis

Real-time qPCR was performed to detect the levels of ferroptosis-related genes. Total RNA was extracted from cells using a total RNA rapid extraction kit (Beyotime, China). Next, cDNA was synthesized from the RNA with a cDNA synthesis kit (ES Science, China). The quantity of the mRNA was measured using a Super SYBR qPCR Master Mix kit (ES Science, China) and was performed in an ABI Prism 7300 real-time thermocycler (Applied Biosystems, Foster City, CA, USA). GADPH was used as an internal reference. Results were calculated using the $2^{-\Delta\Delta Ct}$ method. The primers sequence are shown in Table 1.

### Western blot analysis

Western blot was used to detect the expressions of ferroptosis-related proteins. After the interventional cells were collected, a cell protein lysate (Solarbio, Beijing, China) containing PMSF (Solarbio, Beijing, China) and a phosphatase inhibitor (CWBIO, Beijing, China) was added into the cells to extract cellular proteins, and BCA protein detection kit (Beyotime, Shanghai, China) was used to detect protein concentration. The protein samples were mixed with 4 × loading buffer at a volume of 3:1, and denatured by boiling at 100˚C for 8 min. After electrophoresis on SDS-PAGE gel (10%), the proteins was transferred to PVDF membranes (ISEQ00010; Millipore, Billerica, MA, USA). Subsequently, the membranes were blocked with 5% nonfat dry milk for 2 h and then incubated with p-Nrf2, Nrf2, xCT and GPX4 antibody (1: 1 000) at 4 ˚C overnight. After washed with TBST, the membranes were incubated with the anti-rabbit IgG (H+L) secondary antibodies (1: 5 000) at room temperature for 1 h. The membranes were detected using Infrared dual-color fluorescence imaging detection system (LI-COR Odyssey CLx). GADPH is used as an internal reference.

### Statistical analysis

SPSS 20.0 Statistical software was used for data analysis. Measurement data were expressed as mean ± standard deviation (n = 3). One-way analysis of variance (ANOVA) was performed to compare the differences among multiple groups and $P<0.05$ was deemed a statistically significant difference.

## Results

### Effects of HBO on viability in HT22 cells and PC12 cells

The CCK-8 assay showed that HBO enhances the viabilities of oxygen-glucose deprivation-reperfusion-injured HT22 cells and PC12 cells (Fig 1). Compared with the control group, the

A

B

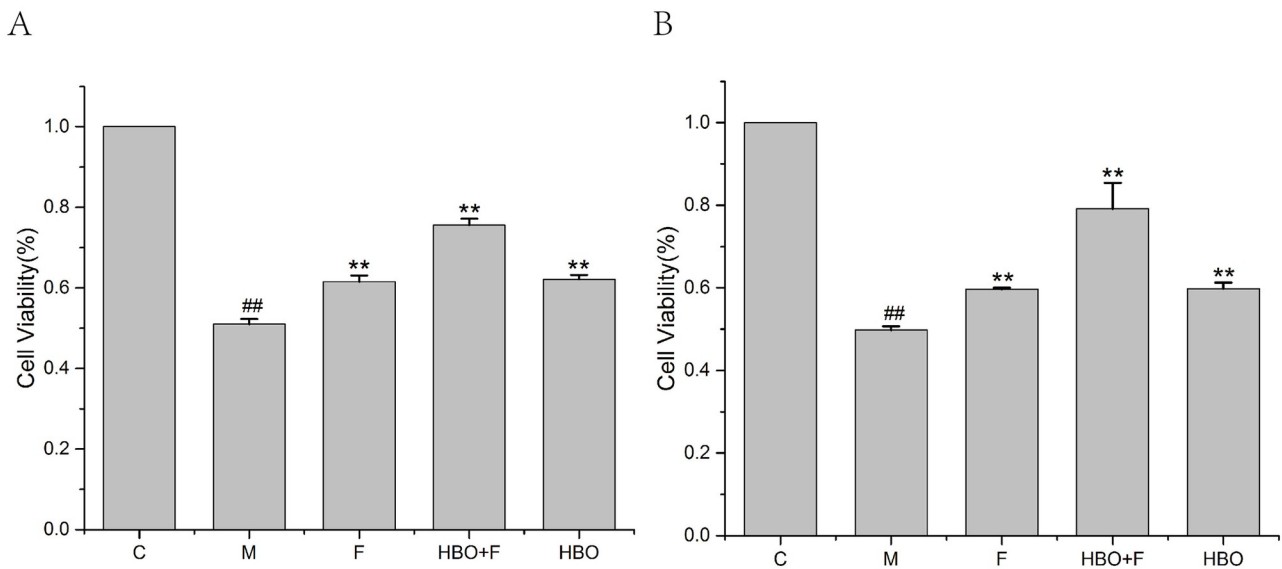

**Fig 1. The effect of HBO on the viability of HT22 cells (A) and PC12 cells (B).** The results are presented as the mean ± SEM (N = 3). ##P < 0.01 vs. control group, **P < 0.01 vs. model group. C: Control group; M: Model group; HBO: Hyperbaric oxygen group; F: Ferrostatin-1 group; HBO+F: Hyperbaric oxygen + Ferrostatin-1 group.

activities of HT22 cells and PC12 cells in the model group were significantly reduced. The activities of HT22 cells and PC12 cells in the HBO group and HBO+F group were significantly higher than the model group.

## Effects of HBO on iron content in HT22 cells and PC12 cells

The iron ion detection experiment showed that compared with the control group, iron content accumulation occurred in HT22 cells and PC12 cells of the model group, while the iron ion content in the cells of the HBO group and HBO+F group was lower than that of the model group. The results showed that HBO could reverse oxygen-glucose deprivation-reperfusion-injured-induced increase in intracellular iron content (Fig 2).

## Effects of HBO on ROS and Lip-ROS in HT22 cells and PC12 cells

Fluorescent probe assay showed that compared with the control group, the ROS and Lipid ROS in the HT22 cells and PC12 cells in the model group were significantly increased, while the levels of ROS and Lipid ROS in the HBO group and HBO+F group were significantly lower than those in the model group. The result suggested that HBO can inhibit the generation of ROS and Lipid ROS in cells (Figs 3 and 4).

## Effects of HBO on MDA and GSH Content in HT22 cells and PC12 cells

The experimental results showed that compared with the control group, the content of DMA in HT22 cells and PC12 cells in the model group was significantly increased, and the content of GSH in the cells was significantly decreased. After HBO treatment, compared with the model group, the content of MDA in the cells was significantly decreased, and the content of GSH in the cells was significantly increased (Fig 5).

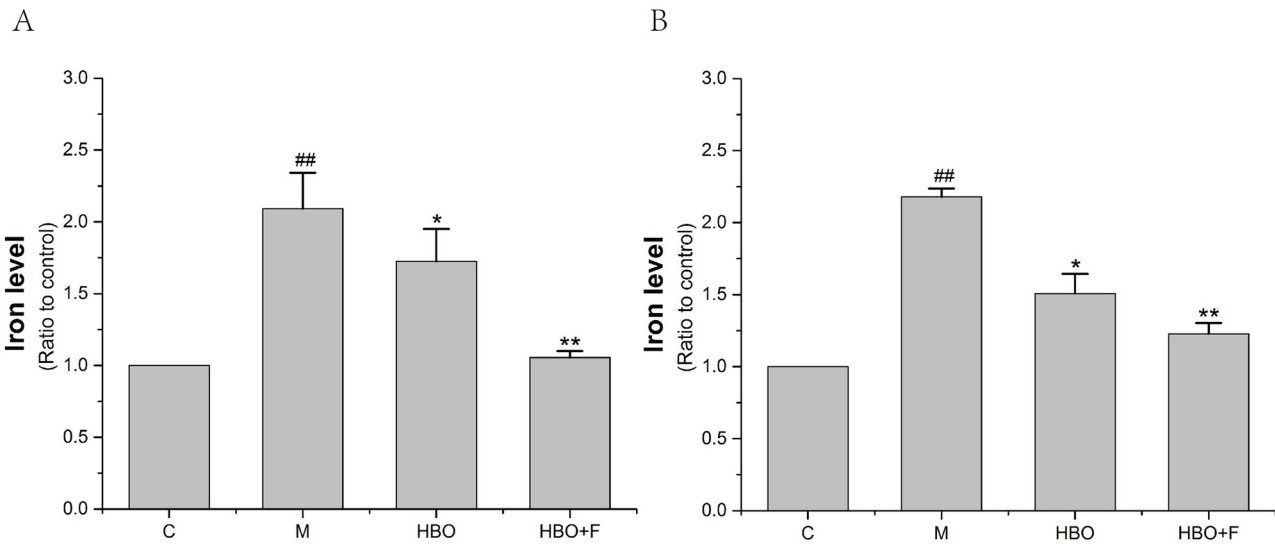

**Fig 2. The effect of HBO on iron content of HT22 cells (A) and PC12 cells (B).** The results are presented as the mean ± SEM (N = 3). ##P < 0.01 vs. control group, *P < 0.05, **P < 0.01 vs. model group. C: Control group; M: Model group; HBO: Hyperbaric oxygen group; HBO+F: Hyperbaric oxygen + Ferrostatin-1 group.

## Effects of HBO on the ultrastructure of ferroptosis in HT22 cells and PC12 cells

The TEM results showed that compared with the control group, the cell membranes of HT22 cells and PC12 cells in the model group were ruptured and blebbed, mitochondria atrophied and became smaller, mitochondrial ridges were reduced or even disappeared, and apoptotic bodies appeared. Compared with the model group, the HT22 cells and PC12 cells in the HBO group and HBO+F group were significantly improved (Figs 6 and 7).

## Effects of HBO on the expression levels of ferroptosis-related genes

PCR results showed that compared with the control group, the expression levels of Nrf2, SLC7A11 and GPX4 genes in the HT22 cells and PC12 cells of the model group were significantly decreased; while the gene expressions of Nrf2, SLC7A11 and GPX4 genes in the cells of HBO group and HBO+F group were significantly higher than those in the model group (Figs 8 and 9).

## Effects of HBO on the expression levels of ferroptosis-related proteins

WB results showed that compared with the control group, the protein expression levels of p-Nrf2/Nrf2, xCT and GPX4 proteins in the HT22 cells of the model group were significantly decreased and the expression levels of p-Nrf2/Nrf2, xCT and GPX4 proteins in the cells of HBO group and HBO+F group were significantly higher than that in the model group (Fig 10).

## Discussion

The process of ferroptosis is accompanied by the accumulation of a large amount of iron ions, and iron is an important cofactor. The balance of iron is essential for the proper functioning of the brain. Studies have found that iron accumulation exists in the brain of patients with many

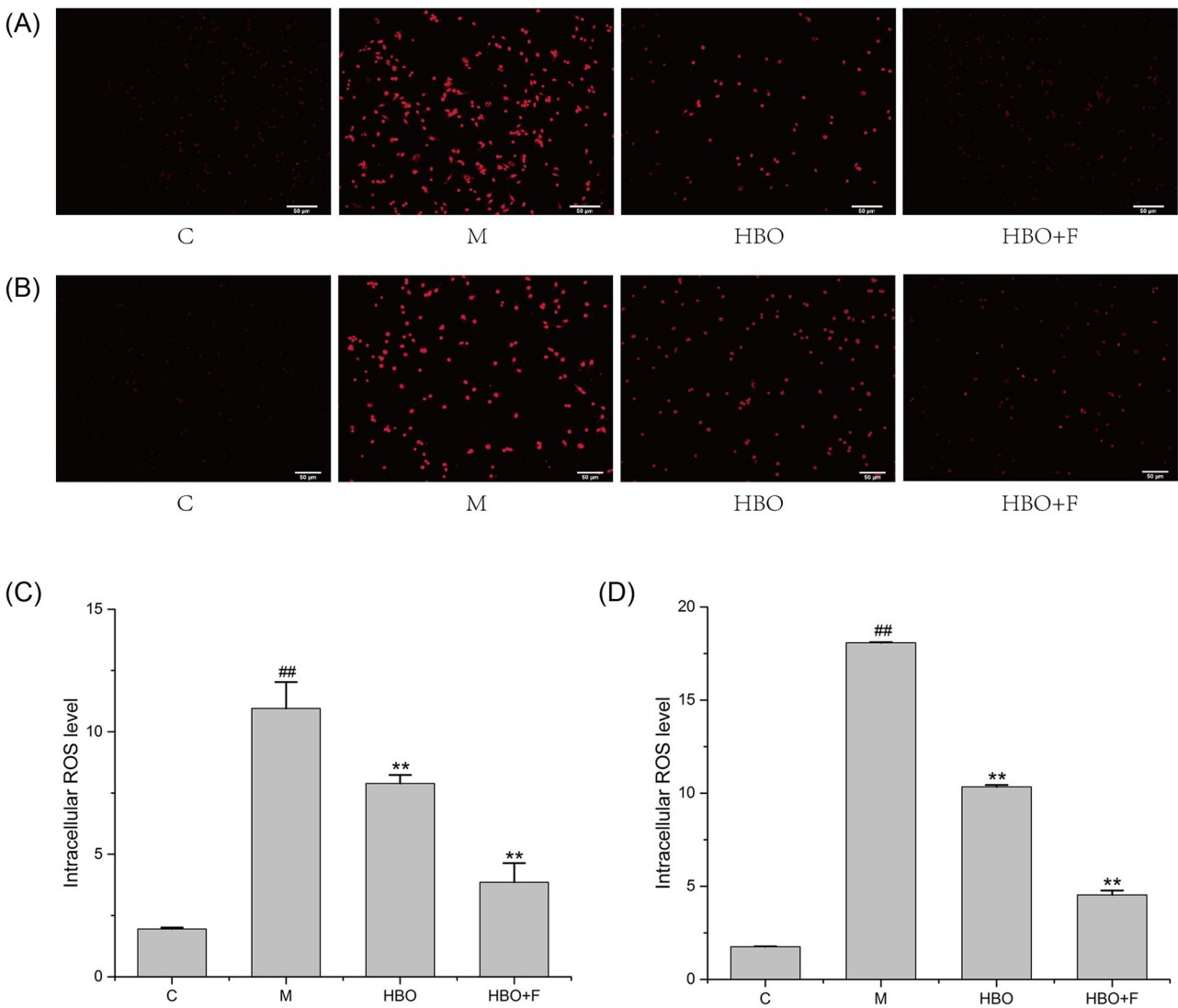

**Fig 3. The effect of HBO on ROS of HT22 cells (A, C) and PC12 cells (B, D) (200×).** The results are presented as the mean ± SEM (N = 3). ##P < 0.01 vs. control group, **P < 0.01 vs. model group. C: Control group; M: Model group; HBO: Hyperbaric oxygen group; HBO+F: Hyperbaric oxygen + Ferrostatin-1 group. ROS: Reactive oxygen species.

neurodegenerative diseases, and iron accumulation can lead to neurotoxicity through various mechanisms, including the production of oxygen free radicals that lead to oxidative stress, excitotoxicity, and promote inflammatory responses [19]. This study found that HBO can significantly reverse the oxygen-glucose deprivation-reperfusion-injury-induced decrease in cell viability and increase in intracellular iron content of HT22 cells and PC12 cells. It has been reported that cerebral ischemia-reperfusion can cause iron metabolism disorders in brain tissue, mainly manifested as increased iron content in brain tissue, resulting in iron overload. The increase of iron content in brain tissue can catalyze the production of free radicals to increase the force, thereby promoting lipid peroxidation and aggravating brain tissue damage [20,21].

Ferroptosis has its unique biochemical characteristics and morphology. Besides iron overload, accumulation of lipid ROS is also a major biochemical feature [22]. Morphological

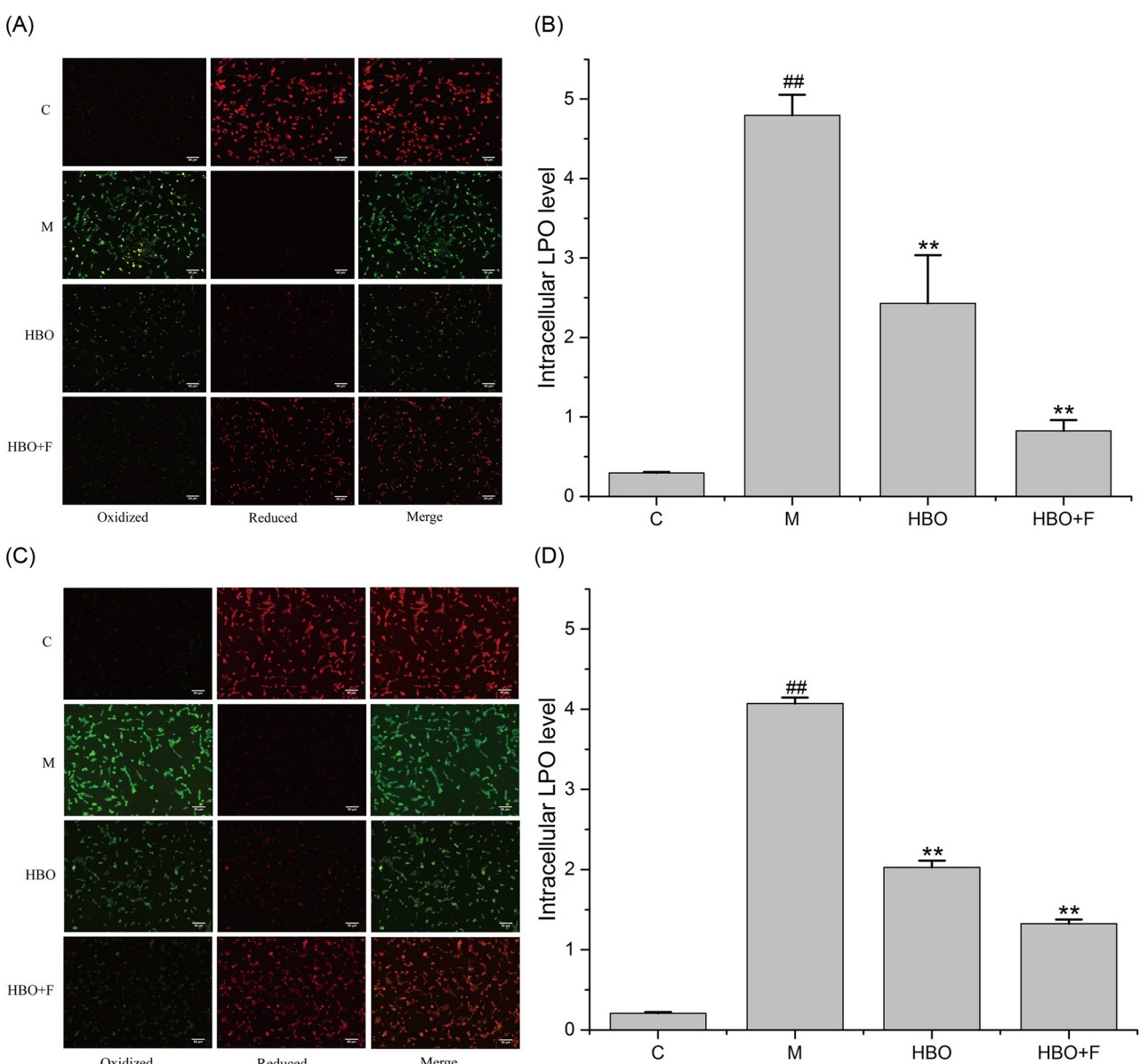

**Fig 4. The effect of HBO on Lipid ROS of HT22 cells (A, B) and PC12 cells (C, D) (200×).** The results are presented as the mean ± SEM (N = 3). ##P < 0.01 vs. control group, *P < 0.05, **P < 0.01 vs. model group. C: Control group; M: Model group; HBO: Hyperbaric oxygen group; HBO+F: Hyperbaric oxygen + Ferrostatin-1 group. LPO: Lipid reactive oxygen species.

features include mitochondrial shrinkage, reduction or disappearance of mitochondrial cristae, increased mitochondrial membrane density, mitochondrial membrane rupture, and normal nuclear morphology, but lack of chromatin condensation [23]. This study showed that ROS, Lip-ROS and MDA were significantly increased, and GSH was significantly decreased in the cells with oxygen-glucose deprivation/reperfusion. TEM showed that the cell membrane was ruptured and blistered, mitochondria atrophied and became smaller, mitochondrial cristae were reduced or even disappeared, and apoptotic bodies appeared in the model groups. And HBO can reverse these changes. The results suggest that oxygen-glucose deprivation/reperfusion can induce ferroptosis in cells, and HBO can inhibit ferroptosis. Iron metabolism

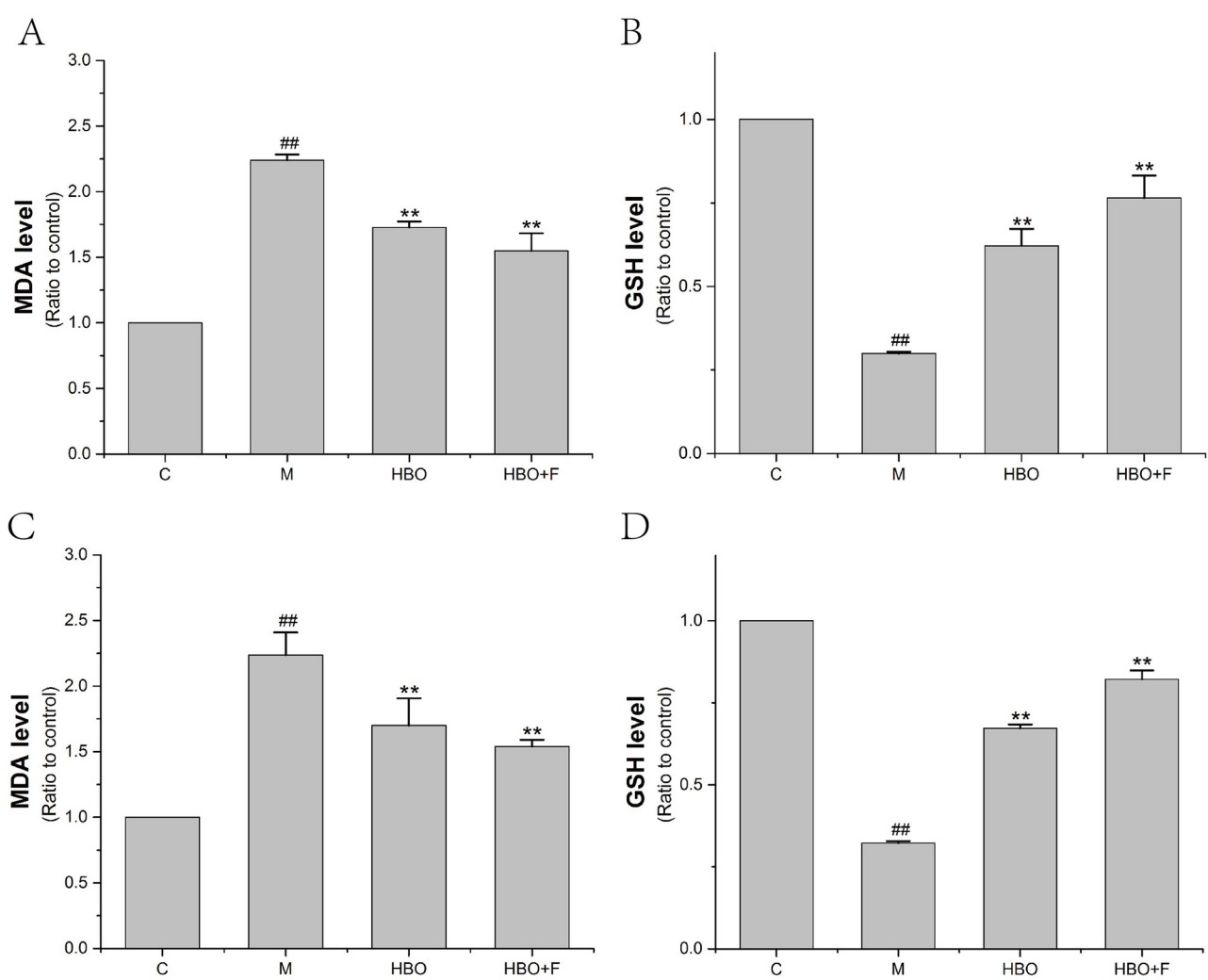

**Fig 5. The effect of MDA and GSH Content of HT22 cells (A, B) and PC12 cells (C, D).** The results are presented as the mean ± SEM (N = 3). ##P < 0.01 vs. control group, **P < 0.01 vs. model group. C: Control group; M: Model group; HBO: Hyperbaric oxygen group; HBO+F: Hyperbaric oxygen + Ferrostatin-1 group.

and lipid peroxidation are key factors mediating ferroptosis. Iron ions in the body are endocytosed into cells through the binding of TFR1 on the cell membrane. Iron ions are reduced to ferrous ions in the cells, and the excessive accumulation of ferrous ions in the cells will promote the generation of oxidative free radicals and lipid peroxidation product MDA through the Fenton reaction, thereby aggravating the damage of nerve cells [24,25].

Combined with our previous and current findings, we found that ferroptosis in the oxygen-glucose deprivation/reperfusion group was accompanied by apoptosis. Apoptosis is programmed series of events dependent on energy, as well as morphological features such as cell shrinkage, chromatin condensation, and presence of apoptotic bodies without inflammatory reactions [26,27]. Caspases are key molecules involved in the transduction of the apoptosis signal, and all of the pathways converge to the executioner caspase-3 [28]. The extrinsic pathway is initiated by the tumor necrosis factor (TNF) receptor family interacting with a ligand and then binds with procaspase-8 following ligand-receptor interaction to activation of caspase-3 which leads to execution of apoptosis [29,30]. The intrinsic pathway (mitochondrial pathway)

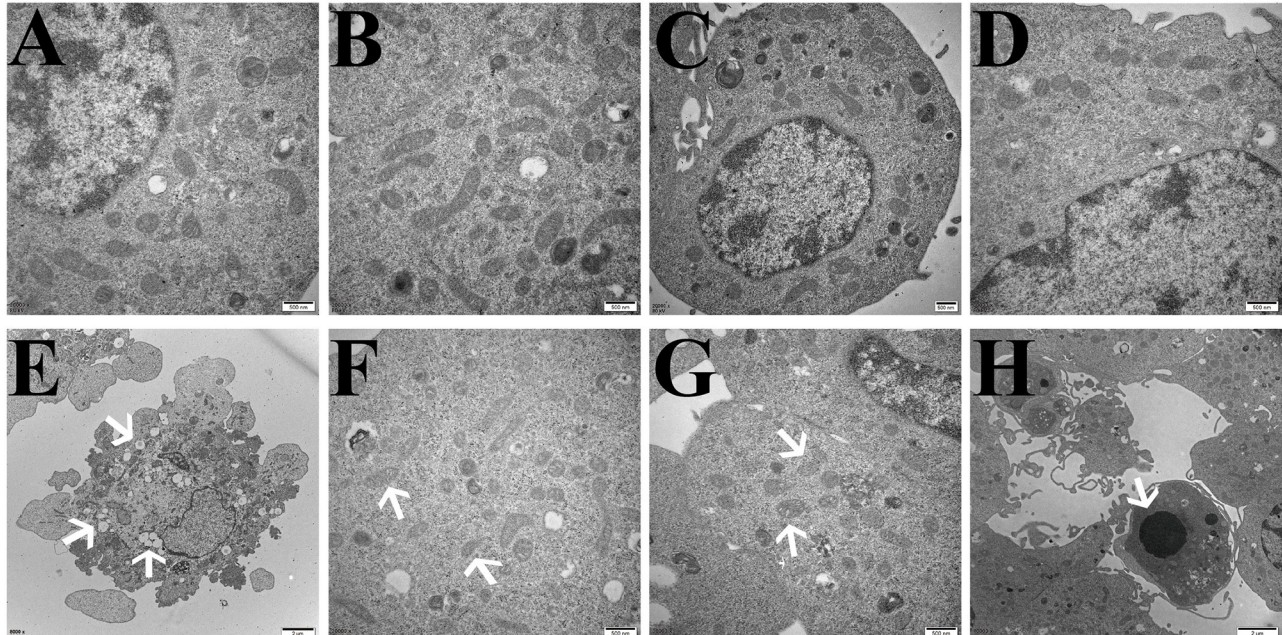

**Fig 6. Ultrastructural changes related to ferroptosis in HT22 cells observed by transmission electron microscopy.** A, B: Control group, with normal cell structure. C, D: HBO group, the cell structure was significantly improved compared with the model group. E,F,G,H: Model group, E: Cell membrane rupture and blebbing; F: Mitochondria become smaller; G: Mitochondrial degeneration; H: Apoptotic body.

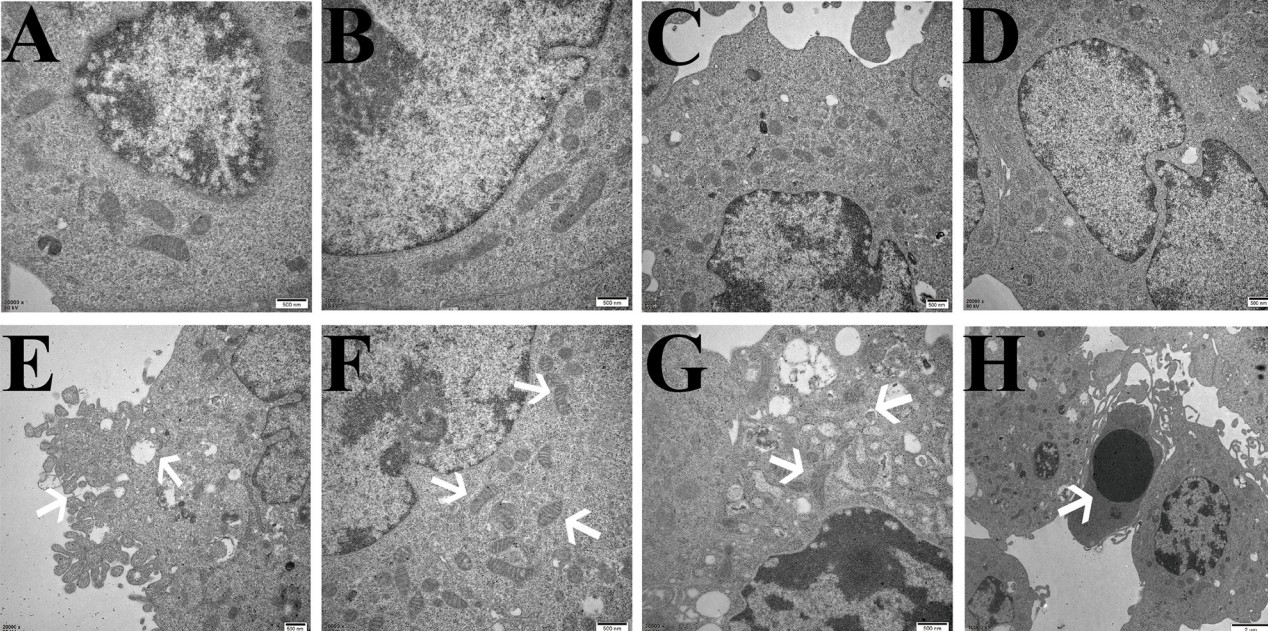

**Fig 7. Ultrastructural changes related to ferroptosis in PC12 cells observed by transmission electron microscopy.** A, B: Control group, with normal cell structure. C, D: HBO group, the cell structure was significantly improved compared with the model group. E,F,G,H: Model group, E: Cell membrane rupture and blebbing; F: Mitochondria become smaller; G: Mitochondrial degeneration; H: Apoptotic body.

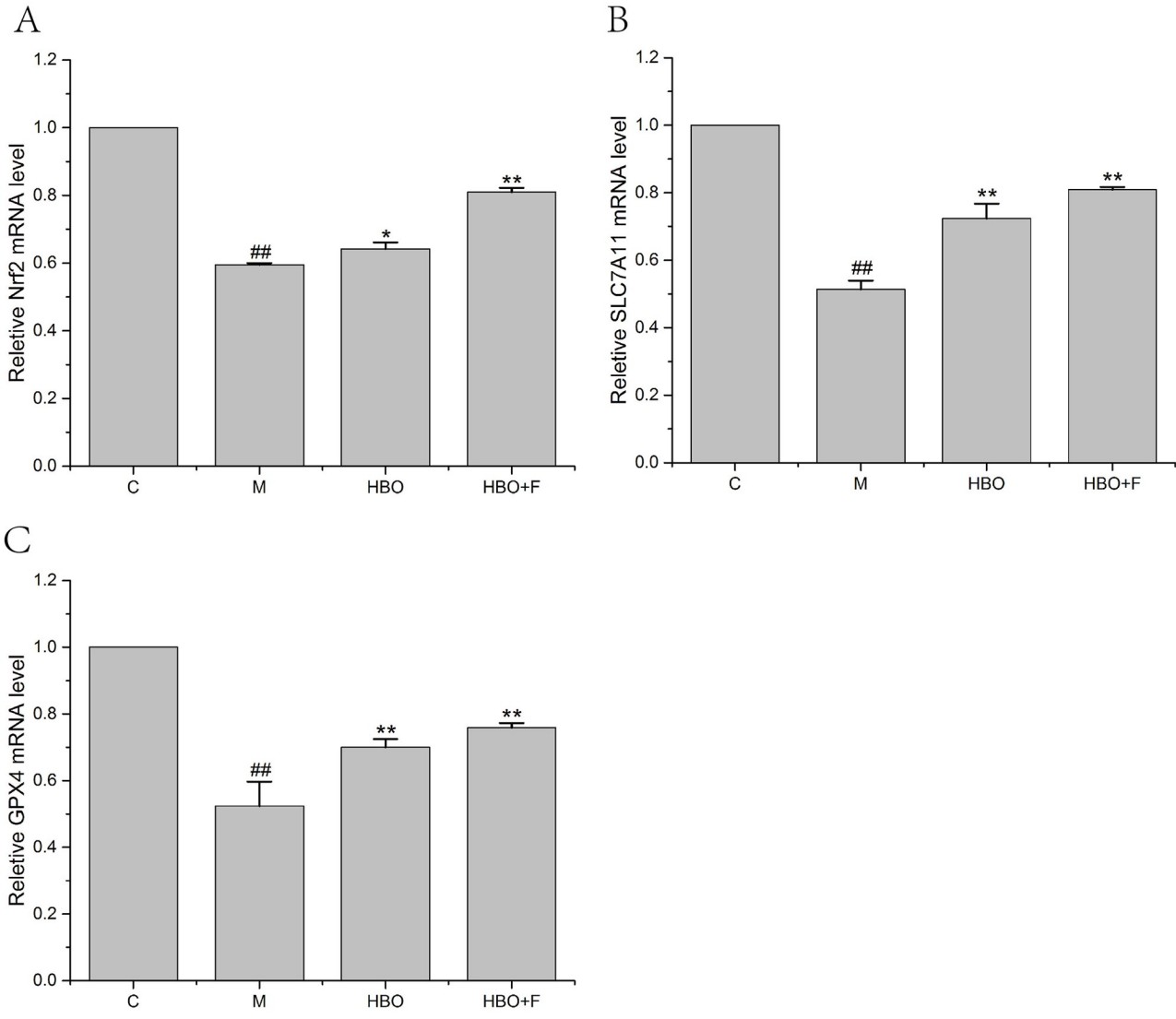

**Fig 8. The effect of HBO on the expression of Nrf2, SLC7A11 and GPX4 genes in HT22 cells.** The results are presented as the mean ± SEM (N = 3). Compared with the control group. ##P < 0.01 vs. control group, *P < 0.05, **P < 0.01 vs. model group. C: Control group; M: Model group; HBO: Hyperbaric oxygen group; HBO+F: Hyperbaric oxygen + Ferrostatin-1 group.

employs alterations of inner mitochondrial membrane for induction of apoptosis. Apoptosis is triggered when the Bcl2-family proapoptotic proteins cause the opening of mitochondrial permeability transition pore and proapoptotic proteins into cytoplasm by interacting with apoptotic protease-activating factor 1 (Apaf-1) and procaspase-9 to constitute apoptosome [31]. An assembly of apoptosome leads to caspase-9 activation, which further activates caspase-3, for apoptotic execution [32]. TEM showed that apoptotic bodies appeared in the model groups. It suggested that apoptosis and ferroptosis coexist under certain conditions. Su et al. [33] found that reactive oxygen species can simultaneously induce apoptosis, autophagy and ferroptosis. Ye et al. [34] found that FBW7-NRA41-SCD1 axis synchronously regulates apoptosis and ferroptosis in pancreatic cancer cells.

As an important intracellular antioxidant molecule, cystine/glutamate transporter (System Xc-) is the upstream node molecule in the process of ferroptosis. Its main function is to

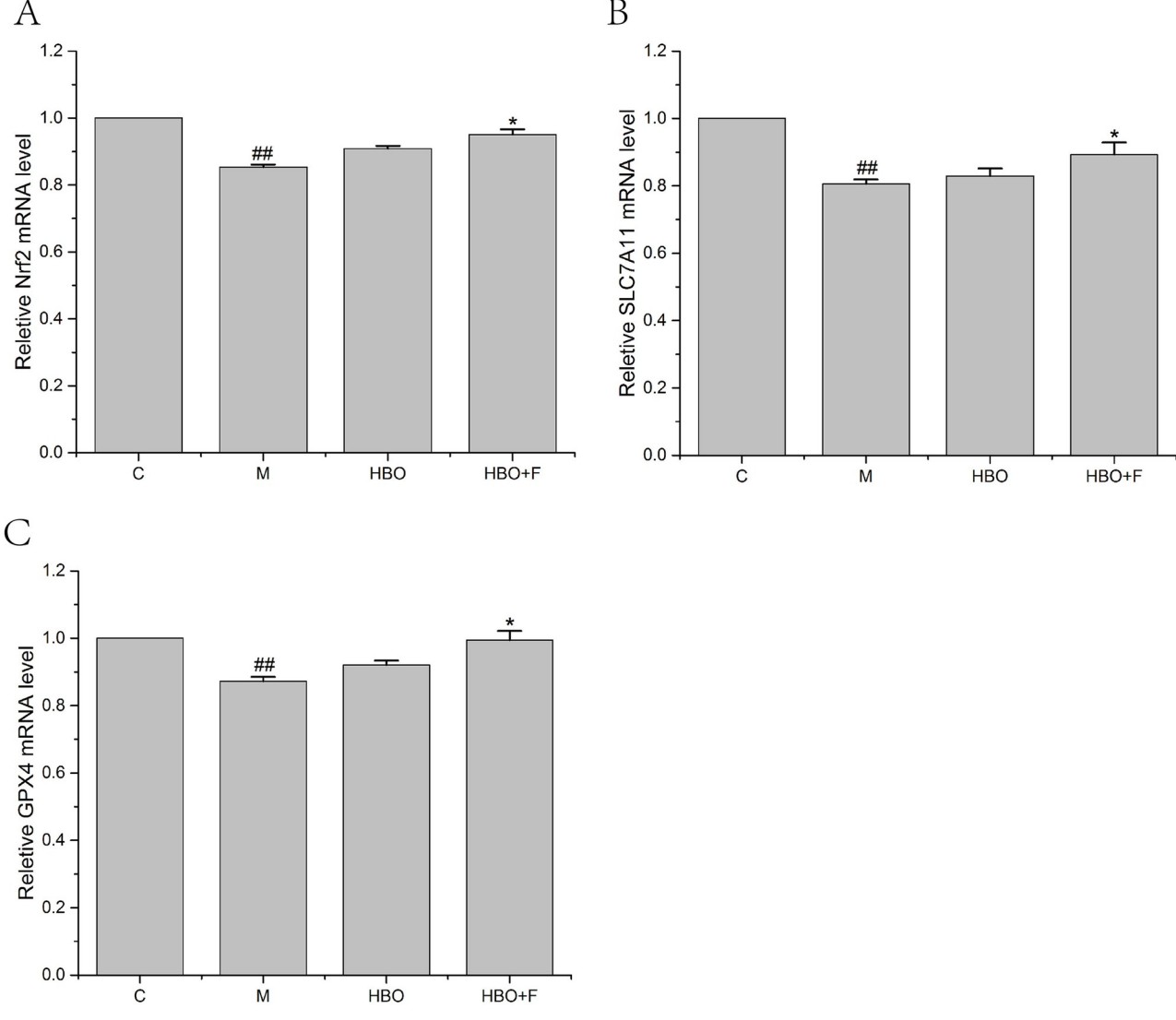

**Fig 9. The effect of HBO on the expression of Nrf2, SLC7A11 and GPX4 genes in PC12 cells.** The results are presented as the mean ± SEM (N = 3). Compared with the control group. ##P < 0.01 vs. control group, *P < 0.05, **P < 0.01 vs. model group. C: Control group; M: Model group; HBO: Hyperbaric oxygen group; HBO+F: Hyperbaric oxygen + Ferrostatin-1 group.

maintain the balance of cystine (Cys) intake and glutamate (Glu) excretion. After being taken up by System Xc-, cystine is reduced to cysteine in cells and is involved in the synthesis of glutathione (GSH). Glutathione can reduce reactive oxygen species and reactive nitrogen species under the action of GPX4 [35]. When System Xc- is inhibited, the introduction of cystine (Cys) into cells is hindered and the cysteine necessary for the synthesis of GSH is reduced. Due to glutathione (GSH) depletion, the activity of glutathione peroxidase 4 (GPX4) is decreased or even inactivated. Intracellular lipid oxides (ROOH) cannot be metabolized to ROH and $H_2O_2$ without oxidative toxicity. The Fenton reaction then occurs, resulting in the production of a large amount of ROS, which severely disrupts the intracellular redox balance, causes cellular lipid peroxidative damage, and attacks biological macromolecules, thereby initiating ferroptosis [36,37]. SLC7A11 (also known as xCT) is the substrate-specific subunit that constitutes System Xc- and is responsible for the transport of cystine from the extracellular to

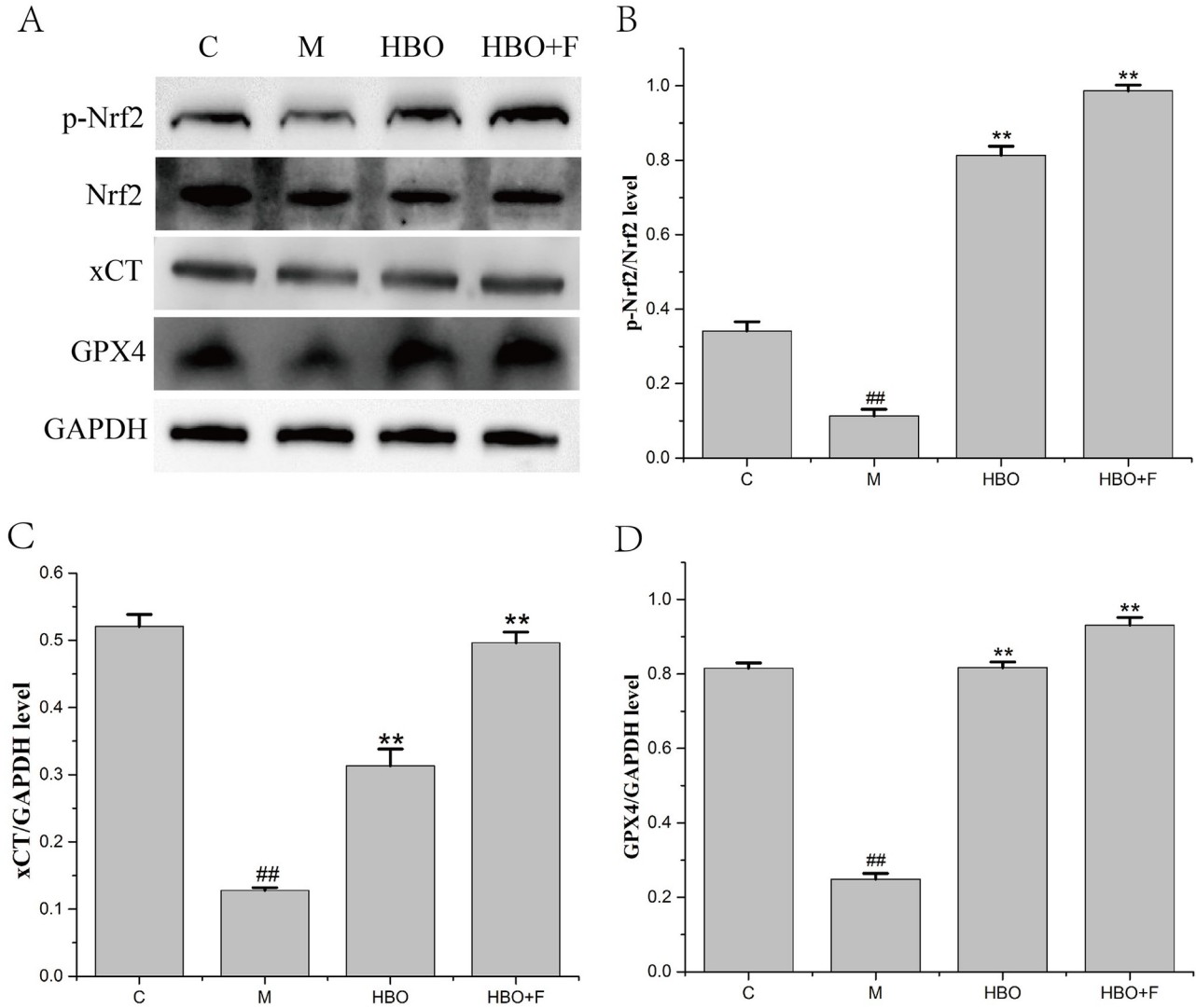

**Fig 10. The effect of HBO on the expression of p-Nrf2/Nrf2, xCT and GPX4 proteins in HT22 cells.** The results are presented as the mean ± SEM (N = 3). Compared with the control group. ##P < 0.01 vs. control group, **P < 0.01 vs. model group. C: Control group; M: Model group; HBO: Hyperbaric oxygen group; HBO+F: Hyperbaric oxygen + Ferrostatin-1 group.

the intracellular. When cells are under oxidative stress and cysteine deficiency, nuclear factor erythroid 2 like 2 (NRF2) and activating transcription factor 4 (ATF4) can induce SLC7A11 expression [38].

The System Xc-/GSH/GPX4 pathway, as one of the main regulatory axes of ferroptosis, plays an important role in cerebral ischemia-reperfusion injury [20,21]. PCR results showed that expressions of Nrf2, SLC7A11 and GPX4 genes were decreased, especially in HT22 cells. WB experiment showed that the expressions of p-Nrf2/Nrf2, xCT and GPX4 proteins in HT22 cells in the model group were significantly decreased. Nrf2-induced decreased SLC7A11 (xCT) expression, decreased glutamate-cysteine exchange, decreased GSH content, decreased GPX4 activity, and insufficient cellular ability to scavenge lipid peroxides resulted in the accumulation of a large number of oxidative products such as MDA. At the mitochondrial membrane, the mitochondrial membrane potential decreases, ferroptosis occurs, and nerve

function is impaired. The expression of these genes and proteins can be reversed after HBO treatment. This suggested that HBO can inhibit ferroptosis through the System Xc-/GSH/ GPX4 pathway.

In summary, our findings indicate that HBO can protect HT22 cells and PC12 cells from damage caused by oxygen-glucose deprivation/reperfusion via the inhibition of Nrf2/System Xc-/GPX4 axis-mediated ferroptosis. Our findings provide a basis for further research on the mechanism of HBO in cerebral ischemia-reperfusion.

## Supporting information

**S1 File.**
(7Z)

**S2 File.**
(7Z)

## Acknowledgments

The authors would like to thank Dr Weidong Li from Guangxi Medical University, China, for his technical support in immunohistochemistry.

## Author Contributions

**Conceptualization:** Chunxia Chen, Xiaorong Pan.

**Data curation:** Wan Chen, Xing Zhou.

**Formal analysis:** Wan Chen, Xing Zhou, Yaoxuan Li.

**Funding acquisition:** Chunxia Chen.

**Investigation:** Yaoxuan Li.

**Methodology:** Xiaorong Pan.

**Writing – original draft:** Wan Chen, Xing Zhou.

**Writing – review & editing:** Chunxia Chen, Xiaorong Pan, Xiaoyu Chen.

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
