## [Decision Letter · Decision Letter 0]

18 May 2022

PONE-D-22-08821Hyperbaric oxygen protects HT22 cells and PC12 cells from damage caused by oxygen-glucose deprivation/reperfusion via the inhibition of Nrf2/System Xc-/GPX4 axis-mediated ferroptosisPLOS ONE

Dear Dr. Chen,

Thank you for submitting your manuscript to PLOS ONE. After careful consideration, we feel that it has merit but does not fully meet PLOS ONE’s publication criteria as it currently stands. Therefore, we invite you to submit a revised version of the manuscript that addresses the points raised during the review process.

We look forward to receiving your revised manuscript.

Kind regards,

Nukhet Aykin-Burns, PhD

Academic Editor

PLOS ONE

Journal Requirements:

3. PLOS requires an ORCID iD for the corresponding author in Editorial Manager on papers submitted after December 6th, 2016. Please ensure that you have an ORCID iD and that it is validated in Editorial Manager. To do this, go to ‘Update my Information’ (in the upper left-hand corner of the main menu), and click on the Fetch/Validate link next to the ORCID field. This will take you to the ORCID site and allow you to create a new iD or authenticate a pre-existing iD in Editorial Manager. Please see the following video for instructions on linking an ORCID iD to your Editorial Manager account: https://www.youtube.com/watch?v=_xcclfuvtxQ.

5. Please ensure that you refer to Figure 10 in your text as, if accepted, production will need this reference to link the reader to the figure.

Reviewers' comments:

Reviewer's Responses to Questions

**Comments to the Author**

1. Is the manuscript technically sound, and do the data support the conclusions?

Reviewer #1: Partly

Reviewer #2: Partly

2. Has the statistical analysis been performed appropriately and rigorously? 

Reviewer #1: N/A

Reviewer #2: N/A

3. Have the authors made all data underlying the findings in their manuscript fully available?

Reviewer #1: No

Reviewer #2: Yes

4. Is the manuscript presented in an intelligible fashion and written in standard English?

Reviewer #1: No

Reviewer #2: Yes

5. Review Comments to the Author

Reviewer #1: The author revealed the neuroprotective effects of hyperbaric oxygen (HBO) in HT22 and PC12 cells, while there are severe problems within the manuscript as listed below:

1. The morphology of PC12 cells is generally round or triangular, and the HT-22 has an elongated/polarized shape. However, the staining results (e.g. Fig 3) demonstrated their PC12 cells did not meet the standard morphology, as we can find that the PC12 here is more of a neuronal-like shape than HT-22. Detail has to be given.

2. The author did not use the standard complete medium (with horse serum) to culture the PC12 cells. Given reasons.

3. “SLC7ALL” (should be SLC7A11) is everywhere throughout the text, while it became correct in the Discussion and part of the figure (Fig 9), which infers the authors did not get the consensus regarding their research target.

4. The Ferrostatin-1 should serve as an individual group to treat the ferroptosis caused by O-G D/R, rather than being administered together with the HBO. Additional experimental results have to be given.

5. Lack of information: the number of repeats for individual experiments; scale bar for Fig3 and Fig4; abbreviations.

Reviewer #2: Major comment:

1. Please clarify whether HBOT protects PC 12 cells from damage caused by oxygen-glucose deprivation/reperfusion via the inhibition of ferroptosis or the inhibition of apoptosis and autophagy (previous publication in BIOCELL, 2022). The authors used the same condition of HBOT to treat PC 12 cells. The previous results showed the protection of HBOT for PC 12 cells from damage caused by oxygen–glucose deprivation/reperfusion via the inhibition of cell apoptosis and autophagy. However, in this study, they showed the protection of HBOT for PC 12 cells from damage caused by oxygen–glucose deprivation/reperfusion via the inhibition of cell ferroptosis. Is it possible that apoptosis, autophagy and ferroptosis can exist at the same condition or the same time, since the morphology, biochemical features and regulatory pathway for all are different?

2. Regarding the pressure and times of HBOT, the authors used 0.25 MPa and 60 min of HBOT to treat the cells. The pressure is higher than that used in clinical treatment. Why and how do the authors choose the proper pressure? If the authors were willing to mimic or simulate the clinical treatment, repetitive treatment of HBOT should be considered. Dose this effect persist after more than one time treatment? Furthermore, the PH value of culture medium may be changed in this condition that could influence the interpretation of protection mechanism.

3. Although the authors checked the morphology, related gene and protein expression of ferroptosis for HBOT treated PC cells directly, there was still some difference between HBO group and HBO+Ferrostatin-1 (HBO+F) group. Did that show statistic difference? It seemed that HBO and ferrostatin-1 had synergistic effect in ferroptosis which meant HBOT protection partially via the inhibition of ferroptosis. The authors may try to pretreat model group with ferrostatin-1 before HBOT to see how it goes. Otherwise, the authors need to justify the title based on the current results.

4. HBOT itself could produce ROS, especially while the higher pressure or the longer duration was used. Although some papers, as the authors cited, indicated HBOT might reduce the oxidative stress in the disease condition, it has been still arguable when HBOT was applied clinically in some neurological disorders, such as stroke. Please describe more to support your view in the discussion section.

Comments on Photos and Figures:

1. Please provide higher resolution images for Figures 3&4

2. Please present the figures differently, such as viability in the curve and intracellular ROS in columns, but combine images and levels of the same cell line with one Fraction.

6. PLOS authors have the option to publish the peer review history of their article (what does this mean?). If published, this will include your full peer review and any attached files.

Reviewer #1: No

Reviewer #2: No

---

## [Author Response · Author response to Decision Letter 0]

16 Aug 2022

Journal Requirements:

3. PLOS requires an ORCID iD for the corresponding author in Editorial Manager on papers submitted after December 6th, 2016. Please ensure that you have an ORCID iD and that it is validated in Editorial Manager. To do this, go to ‘Update my Information’ (in the upper left-hand corner of the main menu), and click on the Fetch/Validate link next to the ORCID field. This will take you to the ORCID site and allow you to create a new iD or authenticate a pre-existing iD in Editorial Manager. Please see the following video for instructions on linking an ORCID iD to your Editorial Manager account: https://www.youtube.com/watch?v=_xcclfuvtxQ.

5. Please ensure that you refer to Figure 10 in your text as, if accepted, production will need this reference to link the reader to the figure.

Author Response: I have addressed the above additional requirements.

Reviewer #1: The author revealed the neuroprotective effects of hyperbaric oxygen (HBO) in HT22 and PC12 cells, while there are severe problems within the manuscript as listed below:

1. The morphology of PC12 cells is generally round or triangular, and the HT-22 has an elongated/polarized shape. However, the staining results (e.g. Fig 3) demonstrated their PC12 cells did not meet the standard morphology, as we can find that the PC12 here is more of a neuronal-like shape than HT-22. Detail has to be given.

Author Response: The PC12 cells in Fig 3 are more numerous, so it appears that the cells has an elongated/polarized shape. In order not to cause misunderstanding, the Fig 3B and Fig 3D have been replaced.

2. The author did not use the standard complete medium (with horse serum) to culture the PC12 cells. Given reasons.

Author Response: Sorry for the wrong description due to my carelessness. PC12 cells were maintained in RPMI 1640 medium supplemented with 10% horse serum, 5% fetal bovine serum and 1% penicillin-streptomycin. Modifications have been made in the Materials and methods section.

3. “SLC7ALL” (should be SLC7A11) is everywhere throughout the text, while it became correct in the Discussion and part of the figure (Fig 9), which infers the authors did not get the consensus regarding their research target.

Author Response: I'm very sorry for miswriting SLC7A11 as SLC7ALL. They have been modified in the manuscript.

4. The Ferrostatin-1 should serve as an individual group to treat the ferroptosis caused by O-G D/R, rather than being administered together with the HBO. Additional experimental results have to be given.

Author Response: A new experiment was added: the cell viability was analyzed when incubated the cells with Ferrostatin-1. As shown in Fig 1C and 1D, like the HBO group, the HT22 and PC12 cell activities of the Ferrostatin-1 groups were significantly higher than those of the model groups. Moreover, PC12 and H9C2 cell activities were higher than those of the HBO groups or the Ferrostatin-1 groups after being treated with Ferrostatin-1 and HBO at the same time. These results indicated that HBO may protect cells from damage by inhibiting ferroptosis.

5. Lack of information: the number of repeats for individual experiments; scale bar for Fig3 and Fig4; abbreviations.

Author Response: The missing information has been added in the manuscript.

Reviewer #2: Major comment:

1. Please clarify whether HBOT protects PC 12 cells from damage caused by oxygen-glucose deprivation/reperfusion via the inhibition of ferroptosis or the inhibition of apoptosis and autophagy (previous publication in BIOCELL, 2022). The authors used the same condition of HBOT to treat PC 12 cells. The previous results showed the protection of HBOT for PC 12 cells from damage caused by oxygen–glucose deprivation/reperfusion via the inhibition of cell apoptosis and autophagy. However, in this study, they showed the protection of HBOT for PC 12 cells from damage caused by oxygen–glucose deprivation/reperfusion via the inhibition of cell ferroptosis. Is it possible that apoptosis, autophagy and ferroptosis can exist at the same condition or the same time, since the morphology, biochemical features and regulatory pathway for all are different?

Author Response: Our findings and literature reports suggested that apoptosis, autophagy, and ferroptosis may coexist. During ferroptosis, the balance of the intracellular redox system is disrupted, leading to mitochondrial damage, which may promote the occurrence of autophagy, which may act as a feedback loop to further induce ferroptosis [1-3]. Autophagy is like a double-edged sword, which can induce apoptosis and inhibit apoptosis [4-6].

1. Li J, Liu J, Xu Y, Wu R, et al. Tumor heterogeneity in autophagy-dependent ferroptosis[J]. Autophagy,2021, 1-14.

2. Liu J, Yang M, Kang R, et al. Autophagic degradation of the circadian clock regulator promotes

ferroptosis[J]. Autophagy, 2019, 15(11): 2033-2035.

3. Munoz P, Casas J, Megias E, et al. The anti-cancer drug ABTL0812 induces ER stress-mediated cytotoxic autophagy by increasing dihydroceramide levels in cancer cells[J]. Autophagy, 2020, 1-18.

4. Maiuri MC, alckvar E, Kimchi A , Kroemer G. Self-eating and self-killing: crosstalk between autophagy and apoptosis. Nat Rev Mol Cell Biol. 2007 Sep;8(9):741-52.

5. Booth LA, Roberts JL, Dent P. The role of cell signaling in the crosstalk between autophagy and apoptosis in the regulation of tumor cell survival in response to sorafenib and neratinib. Semin Cancer Biol. 2020 Nov;66:129-139. 

6. Pang H, Wu T, Peng Z, Tan Q, Peng X, Zhan Z, Song L, Wei B. Baicalin induces apoptosis and autophagy in human osteosarcoma cells by increasing ROS to inhibit PI3K/Akt/mTOR, ERK1/2 and β-catenin signaling pathways. J Bone Oncol. 2022 Feb 1;33:100415. 

2. Regarding the pressure and times of HBOT, the authors used 0.25 MPa and 60 min of HBOT to treat the cells. The pressure is higher than that used in clinical treatment. Why and how do the authors choose the proper pressure? If the authors were willing to mimic or simulate the clinical treatment, repetitive treatment of HBOT should be considered. Dose this effect persist after more than one time treatment? Furthermore, the PH value of culture medium may be changed in this condition that could influence the interpretation of protection mechanism.

Author Response: Guidelines recommend using a pressure of 2 to 3 times standard atmospheric pressure for HBO therapy. Referring to previous studies, the pressure used for HBO treatment in this study was 2.5 times the atmospheric pressure, which did not exceed the recommended safe range. And, the cell viability was detected by CCK8 assay to determine the time of HBO treatment. All experiments in this study have been repeated three times with consistent results. After the pH test paper detection, the pH value of the culture medium basically did not change under the conditions of this study.

3. Although the authors checked the morphology, related gene and protein expression of ferroptosis for HBOT treated PC cells directly, there was still some difference between HBO group and HBO+Ferrostatin-1 (HBO+F) group. Did that show statistic difference? It seemed that HBO and ferrostatin-1 had synergistic effect in ferroptosis which meant HBOT protection partially via the inhibition of ferroptosis. The authors may try to pretreat model group with ferrostatin-1 before HBOT to see how it goes. Otherwise, the authors need to justify the title based on the current results.

Author Response: There was a statistical difference between the HBO group and the HBO+Ferrostatin-1 (HBO+F) group. HBO and ferrostatin-1 have a synergistic effect in inhibiting ferroptosis. A new experiment was added: the cell viability was analyzed when incubated the cells with Ferrostatin-1. As shown in Fig 1C and 1D, like the HBO group, the HT22 and PC12 cell activities of the Ferrostatin-1 groups were significantly higher than those of the model groups. Moreover, PC12 and H9C2 cell activities were higher than those of the HBO groups or the Ferrostatin-1 groups after being treated with Ferrostatin-1 and HBO at the same time. These results indicated that HBO may protect cells from damage by inhibiting ferroptosis.

4. HBOT itself could produce ROS, especially while the higher pressure or the longer duration was used. Although some papers, as the authors cited, indicated HBOT might reduce the oxidative stress in the disease condition, it has been still arguable when HBOT was applied clinically in some neurological disorders, such as stroke. Please describe more to support your view in the discussion section.

Author Response: Hyperbaric oxygen (HBO) therapy is a clinical use of pure oxygen in the environment of 2-3 times atmospheric pressure to treat ischemic and hypoxic diseases, such as carbon monoxide poisoning, cerebral infarction, decompression sickness and coronary heart disease, etc., and has obtained certain curative effects. According to the US FDA and the International Society of Underwater Hyperbaric Medicine, 13 clinical indications including decompression sickness, carbon monoxide poisoning, gas embolism, acute craniocerebral injury, and post-radiotherapy tissue damage can be treated with HBO [1-4]. This part is also highlighted in the introduction part.

1. O. Cheng, R.P. Ostrowski, B. Wu, W. Liu, J.H. Zhang, Cyclooxygenase-2 mediates

hyperbaric oxygen preconditioning in the rat model of transient global cerebral

ischemia, Stroke 42 (2) (2011) 484–490.

2. R.P. Ostrowski, G. Graupner, E. Titova, J. Zhang, J. Chiu, N. Dach, D. Corleone, J. Tang, J.H. Zhang, The hyperbaric oxygen preconditioning-induced brain protection is mediated by a reduction of early apoptosis after transient global cerebral ischemia, Neurobiol. Dis. 29 (1) (2008) 1–13.

3. R.E. Rosenthal, R. Silbergleit, P.R. Hof, Y. Haywood, G. Fiskum, Hyperbaric oxygen

reduces neuronal death and improves neurological outcome after canine cardiac

arrest, Stroke 34 (5) (2003) 1311–1316.

4. Z.N. Guo, L. Xu, Q. Hu, N. Matei, P. Yang, L.S. Tong, Y. He, Z. Guo, J. Tang, Y. Yang,

J.H. Zhang, Hyperbaric oxygen preconditioning attenuates hemorrhagic transformation through reactive oxygen species/thioredoxin-interacting protein/nod-like receptor protein 3 pathway in hyperglycemic middle cerebral artery occlusion rats, Crit. Care Med. 44 (6) (2016) e403–e411.

Comments on Photos and Figures:

1.Please provide higher resolution images for Figures 3&4.

Author Response: Higher resolution images for Figures 3&4 have been provided.

2.Please present the figures differently, such as viability in the curve and intracellular ROS in columns, but combine images and levels of the same cell line with one Fraction.

Author Response: The viability at different times was not researched in this study, so it is not suitable to present it as a curve.

Author Response: Before uploading,PACE have been used to verify the figures.

---

## [Decision Letter · Decision Letter 1]

12 Sep 2022

PONE-D-22-08821R1Hyperbaric oxygen protects HT22 cells and PC12 cells from damage caused by oxygen-glucose deprivation/reperfusion via the inhibition of Nrf2/System Xc-/GPX4 axis-mediated ferroptosisPLOS ONE

Dear Dr. Chen,

Thank you for submitting your manuscript to PLOS ONE. After careful consideration, we feel that it has merit but does not fully meet PLOS ONE’s publication criteria as it currently stands. Therefore, we invite you to submit a revised version of the manuscript that addresses the points raised during the review process. Make sure you answer the critique raised by both reviewers in your revised manuscript as well as in your rebuttal letter, including addition of more relevant references into the paper.

We look forward to receiving your revised manuscript.

Kind regards,

Nukhet Aykin-Burns, PhD

Academic Editor

PLOS ONE

Reviewers' comments:

Reviewer's Responses to Questions

**Comments to the Author**

1. If the authors have adequately addressed your comments raised in a previous round of review and you feel that this manuscript is now acceptable for publication, you may indicate that here to bypass the “Comments to the Author” section, enter your conflict of interest statement in the “Confidential to Editor” section, and submit your "Accept" recommendation.

Reviewer #1: (No Response)

Reviewer #2: (No Response)

2. Is the manuscript technically sound, and do the data support the conclusions?

Reviewer #1: Yes

Reviewer #2: Partly

3. Has the statistical analysis been performed appropriately and rigorously? 

Reviewer #1: Yes

Reviewer #2: No

4. Have the authors made all data underlying the findings in their manuscript fully available?

Reviewer #1: Yes

Reviewer #2: Yes

5. Is the manuscript presented in an intelligible fashion and written in standard English?

Reviewer #1: Yes

Reviewer #2: Yes

6. Review Comments to the Author

Reviewer #1: The author has addressed most concerns after this round revision, though there are minor problems needed to be corrected:

1. Fig 1. The author has added independent Ferrostatin-1 group as shown in Fig1C&D, so the Fig1A&B demonstrating the same experiment should be deleted.

2. “SLC7ALL” (should be SLC7A11) is still existing in Fig 8B.

Reviewer #2: According to the authors’ responses, some problems were not clarified clearly. Please provide more explanation with revised figures to support the rationale.

1) Although more literature suggests that apoptosis, autophagy, and ferroptosis may coexist under certain conditions, please list the percentage of cells undergoing ferroptosis and apoptosis or autophagy in the model (oxygen-glucose deprivation/reperfusion) group. How did you define the apoptotic body in Fig 6H? Was it under the same microscopic field where ferroptosis occurred? What proportion of apoptotic cells was found?

2) As the second comments on photos and figures, please present the figures separately, such as viability in the curve and intracellular ROS in columns, but combine images and levels of the same cell line with one fraction. The reason is that the previous research protocol demonstrated the HBOT protective effect against apoptosis and autophagy by using 0.25 MPa for 90 min/day, however, in the current study against ferroptosis HBOT with 0.25 MPa for 60 min/day was used. Were there the different results in cell viability and ROS levels? How did the authors design the two protocols?

3) In the discussion, please explain the potential differential signaling pathways triggering apoptosis or/and ferroptosis in the model (oxygen-glucose deprivation/reperfusion) group. Because some of listed references were not so relevant, please consider a rearrangement and provide more original discoveries on the coexistence of ferroptosis and apoptotic bodies under the same condition. It may further benefit and lighten the application of HBOT in neurological disorders.

4) The resolution for Figures 3 and 4 remained the same. Please try to treat cells at a proper density (2.5 x105 cells/well) so that you may get a better picture. Some protocols recommend a short-term serum-free DMEM before ROS detection.

7. PLOS authors have the option to publish the peer review history of their article (what does this mean?). If published, this will include your full peer review and any attached files.

Reviewer #1: No

Reviewer #2: No

---

## [Author Response · Author response to Decision Letter 1]

15 Sep 2022

Reviewer #1: The author has addressed most concerns after this round revision, though there are minor problems needed to be corrected:

1. Fig 1. The author has added independent Ferrostatin-1 group as shown in Fig1C&D, so the Fig1A&B demonstrating the same experiment should be deleted.

Author Response: Fig1A&B have been deleted.

2. “SLC7ALL” (should be SLC7A11) is still existing in Fig 8B.

Author Response: It has been revised in Fig 8B.

Reviewer #2: According to the authors’ responses, some problems were not clarified clearly. Please provide more explanation with revised figures to support the rationale.

1) Although more literature suggests that apoptosis, autophagy, and ferroptosis may coexist under certain conditions, please list the percentage of cells undergoing ferroptosis and apoptosis or autophagy in the model (oxygen-glucose deprivation/reperfusion) group. How did you define the apoptotic body in Fig 6H? Was it under the same microscopic field where ferroptosis occurred? What proportion of apoptotic cells was found?

Author Response: Autophagy, apoptosis and ferroptosis exist in the process of cerebral ischemia-reperfusion injury. The formation of apoptotic bodies is because mitochondria, endoplasmic reticulum and other organelles in apoptotic cells are encapsulated by endoplasmic reticulum membrane together with other cytoplasmic components. After autophagosomes fuse with apoptotic cell membranes, autophagosomes are excreted out of cells to become apoptotic bodies. Our preliminary experimental results found that apoptosis and autophagy mainly occurred in oxygen-glucose deprivation for 0.5 h and reperfusion for 24h, and ferroptosis mainly occurred in oxygen-glucose deprivation for 2 h and reperfusion for 24h. This study focused on ferroptosis and did not observe the percentages of apoptosis and autophagy.

2) As the second comments on photos and figures, please present the figures separately, such as viability in the curve and intracellular ROS in columns, but combine images and levels of the same cell line with one fraction. The reason is that the previous research protocol demonstrated the HBOT protective effect against apoptosis and autophagy by using 0.25 MPa for 90 min/day, however, in the current study against ferroptosis HBOT with 0.25 MPa for 60 min/day was used. Were there the different results in cell viability and ROS levels? How did the authors design the two protocols?

Author Response: Literatures show that there is a time window for hyperbaric oxygen therapy, from 60-120min. In the pre-experiment for the current study, HBO with 0.25 MPa for 60, 90 min/day was used separately. It was found that the cell viability and ROS levels of 60 and 90 min/day were the same, so HBO with 0.25 MPa for 60 min/day was selected for this study. 

3) In the discussion, please explain the potential differential signaling pathways triggering apoptosis or/and ferroptosis in the model (oxygen-glucose deprivation/reperfusion) group. Because some of listed references were not so relevant, please consider a rearrangement and provide more original discoveries on the coexistence of ferroptosis and apoptotic bodies under the same condition. It may further benefit and lighten the application of HBOT in neurological disorders.

Author Response: Combined with our previous and current findings, we found that ferroptosis in the oxygen-glucose deprivation/reperfusion group was accompanied by apoptosis. Potential differential signaling pathways triggering apoptosis and ferroptosis in the model (oxy-glucose deprivation/reperfusion) group have been explained in the discussion.

4) The resolution for Figures 3 and 4 remained the same. Please try to treat cells at a proper density (2.5 x105 cells/well) so that you may get a better picture. Some protocols recommend a short-term serum-free DMEM before ROS detection.

Author Response: We tried to get the better pictures for Figures 3 and 4. Figure 3 with higher resolution has been re-uploaded. The re-uploaded Figure 4 is considered higher resolution than the first upload.

---

## [Decision Letter · Decision Letter 2]

28 Sep 2022

Hyperbaric oxygen protects HT22 cells and PC12 cells from damage caused by oxygen-glucose deprivation/reperfusion via the inhibition of Nrf2/System Xc-/GPX4 axis-mediated ferroptosis

PONE-D-22-08821R2

Dear Dr. Chen,

We’re pleased to inform you that your manuscript has been judged scientifically suitable for publication and will be formally accepted for publication once it meets all outstanding technical requirements.

Kind regards,

Yoshiaki Tsuji

Academic Editor

PLOS ONE

Additional Editor Comments (optional):

Reviewers' comments:

Reviewer's Responses to Questions

**Comments to the Author**

1. If the authors have adequately addressed your comments raised in a previous round of review and you feel that this manuscript is now acceptable for publication, you may indicate that here to bypass the “Comments to the Author” section, enter your conflict of interest statement in the “Confidential to Editor” section, and submit your "Accept" recommendation.

Reviewer #1: All comments have been addressed

Reviewer #2: All comments have been addressed

2. Is the manuscript technically sound, and do the data support the conclusions?

Reviewer #1: Yes

Reviewer #2: Yes

3. Has the statistical analysis been performed appropriately and rigorously? 

Reviewer #1: (No Response)

Reviewer #2: Yes

4. Have the authors made all data underlying the findings in their manuscript fully available?

Reviewer #1: Yes

Reviewer #2: Yes

5. Is the manuscript presented in an intelligible fashion and written in standard English?

Reviewer #1: (No Response)

Reviewer #2: Yes

6. Review Comments to the Author

Reviewer #1: The author has addressed all the concerns after this round of revision.There is no further questions.

Reviewer #2: (No Response)

7. PLOS authors have the option to publish the peer review history of their article (what does this mean?). If published, this will include your full peer review and any attached files.

Reviewer #1: No

Reviewer #2: No

---

## [Editor Report · Acceptance letter]

26 Oct 2022

PONE-D-22-08821R2 

Hyperbaric oxygen protects HT22 cells and PC12 cells from damage caused by oxygen-glucose deprivation/reperfusion via the inhibition of Nrf2/System Xc-/GPX4 axis-mediated ferroptosis 

Dear Dr. Chen:

I'm pleased to inform you that your manuscript has been deemed suitable for publication in PLOS ONE. Congratulations! Your manuscript is now with our production department. 

Kind regards, 

on behalf of

Dr. Yoshiaki Tsuji 

Academic Editor

PLOS ONE